# ADAP1 promotes invasive squamous cell carcinoma progression and predicts patient survival

Avery Van Duzer[1,]*, Sachiko Taniguchi[1,]*, Ajit Elhance[1], Takahiro Tsujikawa[1,3], Naoki Oshimori[1,2,3,4]

Invasive squamous cell carcinoma (SCC) is aggressive cancer with a high risk of recurrence and metastasis, but the critical determinants of its progression remain elusive. Here, we identify ADAP1, a GTPase-activating protein (GAP) for ARF6 up-regulated in TGF-β-responding invasive tumor cells, as a strong predictor of poor survival in early-stage SCC patients. Using a mouse model of SCC, we show that ADAP1 overexpression promotes invasive tumor progression by facilitating cell migration and breakdown of the basement membrane. We found that ADAP1-rich, TGF-β-responding tumor cells exhibit cytoplasmic laminin localization, which correlated with the absence of laminin and type IV collagen from the pericellular basement membrane. Interestingly, although tumors overexpressing a GAP activity-deficient mutant of ADAP1 resulted in morphologically complex tumors, those tumor cells failed to breach the basement membrane. Moreover, *Adap1* deletion in tumor cells ameliorated the basement membrane breakdown and had less invading cells in the stroma. Our study demonstrates that ADAP1 is a critical mediator of TGF-β-induced cancer invasion and might be exploited for the treatment of high-risk SCC.

## Introduction

Invasive squamous cell carcinomas (SCCs) arising from skin, lung, oral, esophagus, and cervical epithelial tissues are significant contributors to cancer mortality worldwide (Dotto & Rustgi, 2016). SCC is genetically and molecularly heterogeneous, which makes it challenging to identify the relatively rare, high-risk SCCs that may progress to life-threatening malignancies. Only 5–20% of cutaneous SCC cases progress to regional metastasis (Alam & Ratner, 2001; Moore et al, 2005; Kang & Toland, 2016), but of these cases, the 5-yr survival rate is only 25–35% (Rowe et al, 1992; Kraus et al, 1998). It is, therefore, essential to find molecular targets unique to metastatic cases to diagnose and treat high-risk SCCs effectively.

Both tumor cell-intrinsic and tumor cell-extrinsic factors can activate molecular pathways that promote invasive tumor growth and metastasis, including actomyosin-based cell motility and breakdown of the basement membrane (BM) (Hamidi & Ivaska, 2018). In particular, TGF-β plays multiple roles in cancer invasion and metastasis (David & Massagué, 2018). To study the role of TGF-β in tumor development, we previously developed a mouse model of SCC that harbors an in vivo fluorescent reporter and lineage tracing system for the TGF-β–SMAD2/3 signaling pathway (Oshimori et al, 2015). Using this system, we showed that TGF-β-responding tumor cells are drug-resistant, stem-like tumor-initiating cells (TICs) that promote invasive tumor growth. Therefore, the mechanisms by which TGF-β-responding TICs acquire invasive properties may be a potential target for novel cancer diagnostics and treatment.

Here, we search prognostic genes of SCC from the list of up-regulated genes in TGF-β-responding TICs by in silico analysis. We identify ADAP1 (ArfGAP with dual pleckstrin homology domains 1, also known as centaurin-α1) as a strong predictor of poor survival in early-stage SCC patients. ADAP1 was originally identified as a neuron-specific phosphatidylinositol 3,4,5-trisphosphate (PIP$_3$) and inositol 1,3,4,5-tetrakisphosphate (IP$_4$)-binding protein (Hammonds-Odie et al, 1996; Kreutz et al, 1997) and is involved in dendrite branching and spine development (Moore et al, 2007). ADAP1 has an N-terminal zinc finger ArfGAP domain, which facilitates the activity of the small GTPase ADP-ribosylation factor 6 (ARF6) to hydrolyze GTP to GDP (Thacker et al, 2004; Venkateswarlu et al, 2004). Importantly, it is known that ARF family proteins do not have detectable intrinsic GTPase activity (Randazzo & Kahn, 1994; Klein et al, 2006), and thus, GTPase-activating proteins (GAPs), such as ADAP1, are crucial for ARF function.

ARF6 is an essential regulator of endocytic membrane trafficking and is involved in the internalization and externalization of various membrane proteins, including growth factor receptors, integrins, and membrane-type matrix metalloproteases (Marchesin et al, 2015; Charles et al, 2016; Osmani et al, 2018). ARF6 and its regulators have been implicated in tumor development and metastasis (Hashimoto et al, 2004; D'Souza-Schorey & Chavrier, 2006). However, the role of ADAP1 in tumorigenesis and its contribution to ARF6-mediated tumor progression had not previously been evaluated. In

[1]Department of Cell, Developmental & Cancer Biology, Oregon Health and Science University, Portland, OR, USA [2]Department of Dermatology, Oregon Health and Science University, Portland, OR, USA [3]Department of Otolaryngology, Head & Neck Surgery, Oregon Health and Science University, Portland, OR, USA [4]Knight Cancer Institute, Oregon Health and Science University, Portland, OR, USA

Correspondence: oshimori@ohsu.edu
*Avery Van Duzer and Sachiko Taniguchi contributed equally to this work

comparison with other ArfGAPs, ADAP1 may be of particular importance to cancer progression, as it also has a GAP-independent role in actin cytoskeleton remodeling via its C-terminal dual pleckstrin homology domains (Thacker et al, 2004; Venkateswarlu et al, 2004). Here, we show that ADAP1 facilitates SCC progression through both its GAP activity-dependent and GAP activity-independent functions.

Invasive SCC is characterized by the discontinuity of the BM and the emergence of invading tumor cells in the stroma (Yanofsky et al, 2011). The BM is the sheet-like ECM that underlies epithelial tissues and is composed primarily of type IV collagen and laminin. Laminin binds to the extracellular domain of integrins expressed on the basal side of epithelial cells (e.g., $\alpha6\beta4$ integrins in hemidesmosome) and self-assembles into a cell-associated network, the lamina lucida, which is thought to trigger recruitment of type IV collagen. Type IV collagen forms an independent network, the lamina densa, which interacts with the laminin network through other BM components (Kelley et al, 2014).

In this study, we show that ADAP1 promotes invasive SCC progression by activating both cell migration and invasion. Interestingly, ADAP1-rich, TGF-$\beta$-responding tumor cells exhibited cytoplasmic laminin localization and disrupted the BM of their pericellular regions, which was mitigated by *Adap1* deletion. Moreover, tumors overexpressing the GAP activity-deficient mutant of ADAP1 failed to breach the BM. Together, our study establishes ADAP1 as a potential prognostic marker and provides novel insights into the molecular pathway of invasive tumor progression.

## Results

### ADAP1, a strong predictor of SCC patient survival, is up-regulated in TGF-$\beta$-responding TICs

We previously demonstrated that TGF-$\beta$-responding TICs promote invasive SCC progression and tumor recurrence in a mouse model of SCC (Oshimori et al, 2015). Because gene expression signatures of TICs can be used to predict poor patient outcomes (Liu et al, 2007; Eppert et al, 2011), we sought genes with prognostic value specifically expressed in TGF-$\beta$-responding TICs. To do so, we first narrowed down a list of genes that were significantly up-regulated in TGF-$\beta$-responding tumor cells (Oshimori et al, 2015) by the following criteria: (1) genes showed >2-fold up-regulation in TGF-$\beta$-responding cells compared with nonresponding cells were included (adj $P$ < 0.05); (2) genes had fragments per kilobase million > 5 in TGF-$\beta$-responding cells were included; and (3) the genes expressed less in primary tumors than normal tissue according to The Cancer Genome Atlas (TCGA) head and neck SCC (HNSCC) data were excluded (fold-change < 0.5, adj $P$ < 0.05, Mann–Whitney U-test). Next, from the remaining 292 genes with overall survival data for stage I/II HNSCC cases in the TCGA dataset, we identified genes strongly associated with patient survival through multivariate Cox proportional hazards regression of mRNA expression. This analysis revealed that *ADAP1* had a hazard ratio of 1.59 (lower/upper confidence interval, 1.19/2.11) (Fig S1A–C). Because *Adap1* had the greatest fold up-regulation in TGF-$\beta$-responding tumor cells compared with nonresponding counterparts in our murine model (Fig S1D), we focused on ADAP1 in this study. To evaluate *ADAP1* gene expression as a potential prognostic marker, we divided TCGA patient data into three groups (*ADAP1* low-, mid-, and high-expression groups) and generated Kaplan–Meier curves with the overall survival data. When we analyzed patient data of all stages ($n$ = 519), survival rates significantly decreased as *ADAP1* expression increased ($n$ = 173 for each group) (Fig 1A). Strikingly, this negative correlation was further evident in stage I/II patients ($n$ = 118); the 5-yr survival rate of *ADAP1* high-expressing group was 0% ($n$ = 39), whereas mid- and low-expressing groups were about 50% ($n$ = 39) and 70% ($n$ = 40), respectively (Fig 1B), indicating that *ADAP1* expression might be a powerful predictor of high-risk SCC patients.

In mice, we found ADAP1 protein expression to be undetectable in the normal skin epidermis. This was expected, as ADAP1 was

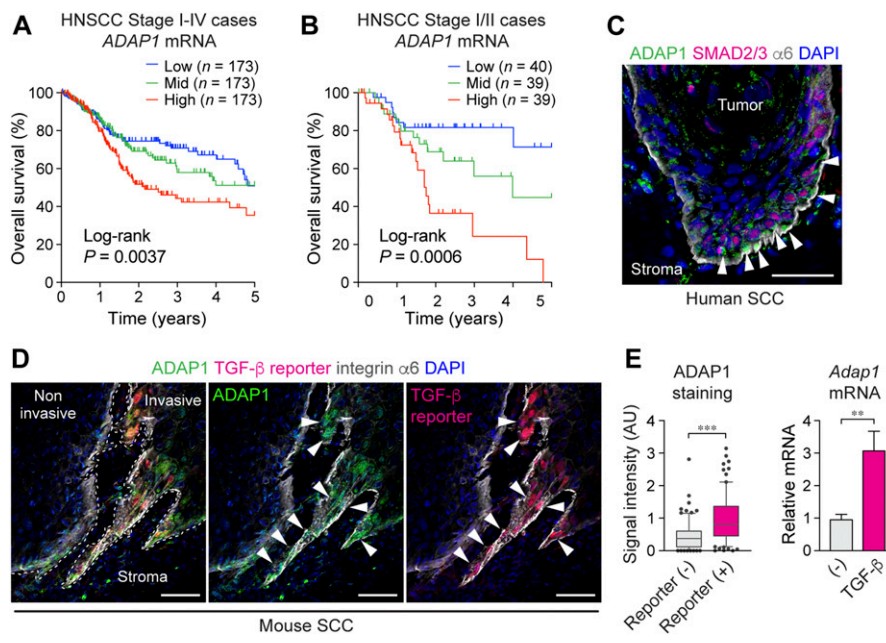

**Figure 1. ADAP1, a strong predictor of SCC patient survival, is up-regulated in TGF-$\beta$-responding TICs.**
**(A, B)** The Kaplan–Meier curves were shown as overall survival of patients with HNSCC in (A) all cases (stage I–IV, $n$ = 519) and (B) early-stage cases (stage I and II, $n$ = 118) (TCGA). Cases were equally stratified into three groups based on *ADAP1* mRNA expression levels, and survival rates of ADAP1 high-, mid-, and low-expression groups were plotted. $P$ value by log-rank (Mantel–Cox) test is indicated. **(C)** Immunolabeling of invasive mouse SCC tumor sections shows increased ADAP1 (green) staining in TGF-$\beta$ reporter⁺ cells (red). Quantification of ADAP1 signal intensity along the leading edge of tumor sections shows significantly increased levels of ADAP1 in TGF-$\beta$ reporter⁺ cells. $n$ = 6. Data are shown in box & whisker plots: mid-line, median; box, 25th and 75th percentiles; whiskers, 5th and 95th percentiles with outliers. ***$P$ < 0.001. $P$ value by unpaired $t$ test is indicated. **(D)** Immunolabeling of human SCC tumor sections increased ADAP1 (green) staining near TGF-$\beta$ signaling high regions, shown via nuclear Smad2/3 staining (red). **(E)** Quantification of relative *Adap1* mRNA expression in primary mouse keratinocytes grown at high cell density, which was untreated or treated with TGF-$\beta$. $n$ = 3. Data are mean ± SD. **$P$ = 0.0043. $P$ value by unpaired $t$ test is indicated. Scale bars, 50 $\mu$m.

reported as a neuron-specific protein (Hammonds-Odie et al, 1996; Kreutz et al, 1997). However, in both mouse and human tumors, we detected ADAP1 protein expression in tumor cells at the leading edges of invasive tumors (Fig 1C and D). Furthermore, in accordance with our RNA-seq data, TGF-β-responding tumor basal cells had significantly stronger cytoplasmic staining of ADAP1 protein (Fig 1C). We observed similar localization patterns of ADAP1 protein in human SCC patient samples, particularly in cells exhibiting nuclear SMAD2/3 localization, which is a sign of active TGF-β signaling (Fig 1D). To examine whether TGF-β plays a role in the up-regulation of *ADAP1* gene expression, we stimulated primary mouse keratinocytes with TGF-β in vitro. *Adap1* mRNA was significantly up-regulated by extended TGF-β treatment in higher cell density culture conditions (Fig 1E), suggesting that TGF-β can induce *ADAP1* gene expression. Although higher cell culture density was not necessary for up-regulating ADAP1 in TGF-β-treated cells, it is possible that other factors induced by higher density may have an additive effect with TGF-β.

## ADAP1 overexpression accelerates cell migration in vitro independent of its GAP activity

To gain insight into the impact of elevated ADAP1 expression, we established a doxycycline (Dox)-inducible ADAP1 overexpression system. We first transduced primary mouse keratinocytes with a lentiviral vector encoding *TetO* promoter-driven nuclear red fluorescent protein (*NLS-mRFP*) and the hygromycin-resistant gene (*HygR*). After hygromycin selection, the cells were further transduced with another lentiviral vector that encodes the transactivator (*rtTA*) and *ADAP1* under the *TetO* promoter (Fig 2A). As a result, in the presence of Dox, rtTA induces the expression of NLS-mRFP and ADAP1. We used the second vector without *ADAP1* as a control (Fig 2B). Interestingly, we found that ADAP1-overexpressing cells exhibit extended cell bodies and long protrusions (Fig 2C), raising the possibility that ADAP1 may promote cell migration and invasion.

To analyze the migration of ADAP1-overexpressing cells, we performed live-cell imaging in vitro. We tracked cellular movement based on the position of fluorescently labeled (NLS-mRFP⁺) nuclei. As shown in Videos 1 and 2, ADAP1-overexpressing cells dynamically formed lamellipodia and long protrusions and migrated strikingly longer distances than control cells expressing only NLS-mRFP. Automated tracking of the NLS-mRFP signal showed that the mean velocity of cell movement was significantly higher in ADAP1-overexpressing cells than in control cells (Fig 2D and E and Videos 3 and 4). To test whether its GTPase-activating function mediates enhanced migratory phenotypes induced by ADAP1 overexpression, we generated a GAP-deficient mutant of ADAP1 by substituting the critical arginine for GAP activity to lysine (R49K) (Fig 2B), based on the structural alignment of the ArfGAP domain (Mandiyan et al, 1999). Live cell imaging showed that cells overexpressing the ADAP1-R49K

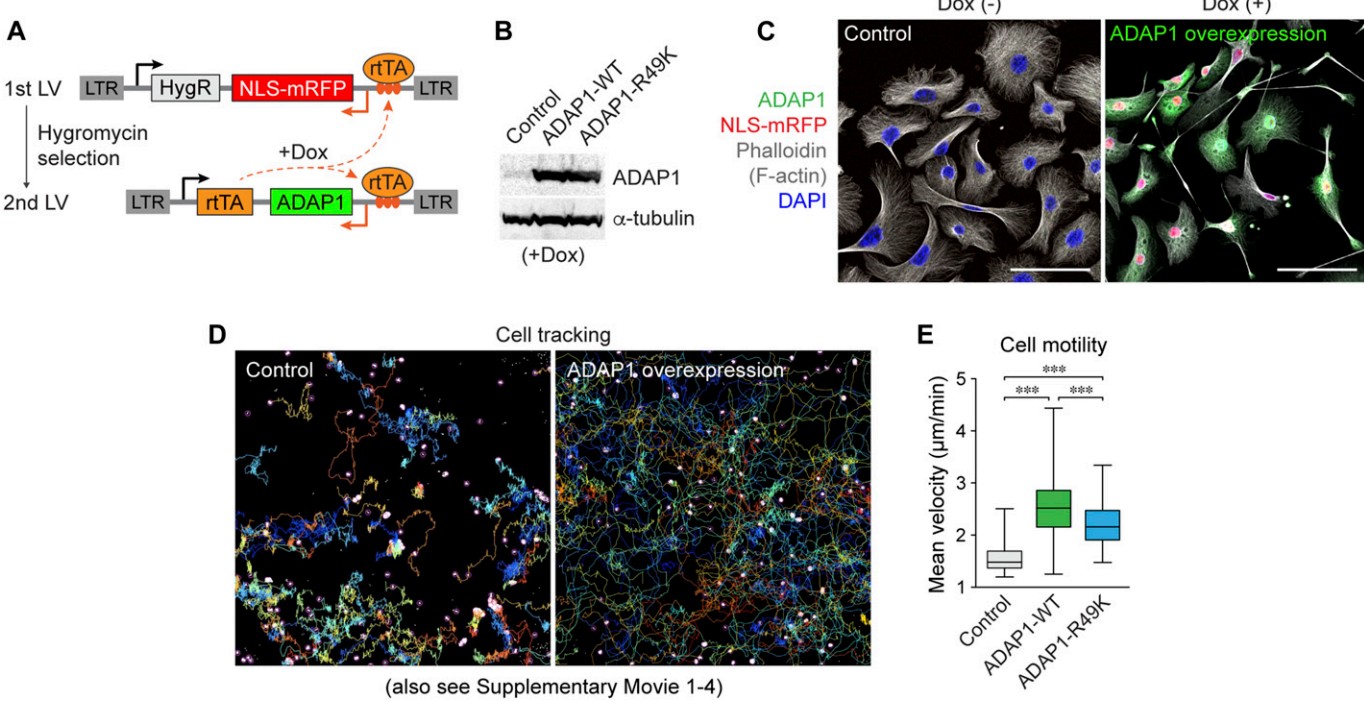

**Figure 2. ADAP1 overexpression accelerates cell migration in vitro independent of its GAP activity.**
**(A)** Lentiviral vector (LV) constructs for doxycycline-induced ADAP1 overexpression. Cells transduced with the first LV were selected by puromycin. Then, the second LV was transduced at <1 MOI, to generate ADAP1-overexpressing cells (NLS-mRFP⁺) and control (NLS-mRFP⁺) cells. **(B)** Western blot analysis showed the overexpression of ADAP1-WT and ADAP1-R49K mutant proteins at similar levels. α-tubulin, loading control. **(C)** Primary mouse keratinocytes transduced with the LVs. Note, doxycycline (Dox)-inducible ADAP1 overexpression (NLS-mRFP⁺ cells) showed extended protrusions and cell elongation. **(D)** Cell tracking images of LV-transduced cells. Nuclear red fluorescent signal (NLS-mRFP) was used to track cell motility (see Videos 1–4), and cell tracking lines are shown. **(D, E)** Quantification of ADAP1-induced cell motility tracking from (D). Note, ADAP1 wild-type (WT) and ADAP1-R49K mutant cells show significantly increased motility compared with control cells, and there is a smaller difference in motility between the two ADAP1 overexpressing groups. *n* = 109–242 cells. Data are shown in box & whisker plots. ***$P$ < 0.001. $P$ value by unpaired $t$ test is indicated. Scale bars, 50 μm.

mutant migrated significantly faster than control cells, although they were slightly, but significantly, slower than cells expressing the wild-type ADAP1 (ADAP1-WT) (Fig 2E). Therefore, the elevated cell motility observed in culture dishes is primarily due to a GAP-independent function of ADAP1 likely through the regulation of actin cytoskeleton rearrangement (Thacker et al, 2004; Venkateswarlu et al, 2004), but the GAP activity may be necessary for even faster cellular movement.

### ADAP1 overexpression promotes cell invasion in vitro via its GAP activity for ARF6

We next assessed if ADAP1 influences the ability to migrate through the ECM by the Matrigel-coated transwell assay. ADAP1-WT overexpression endowed primary keratinocytes with increased invasive

activities compared with control cells (Fig 3A). Despite increased migration, cells overexpressing the ADAP1-R49K mutant failed to increase invasive capability through Matrigel (Fig 3A), suggesting that the GAP activity of ADAP1 is required for enhanced cell invasion. Because ADAP1 stimulates the GTPase activity of ARF6, invasive phenotypes induced by ADAP1 are potentially dependent on ARF6. To test whether ARF6 mediates the pro-invasive function of ADAP1, we depleted *Arf6* mRNA from ADAP1-WT-overexpressing cells by shRNA-mediated knockdown (Fig 3B). ADAP1-induced cell invasion was significantly mitigated by *Arf6* depletion (Fig 3C). Because the intrinsic GTPase activity of ARF proteins is extremely low (Randazzo & Kahn, 1994; Klein et al, 2006), ADAP1 may play a crucial role in facilitating the cycling between ARF6-GTP and ARF6-GDP states. To test this hypothesis, we overexpressed a constitutively

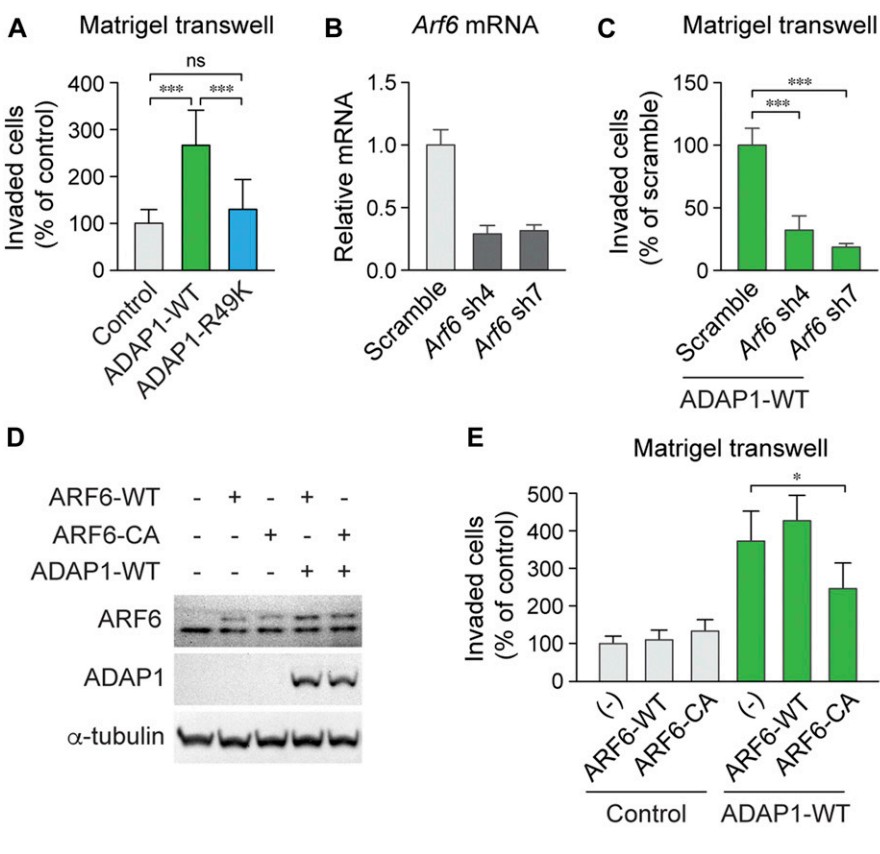

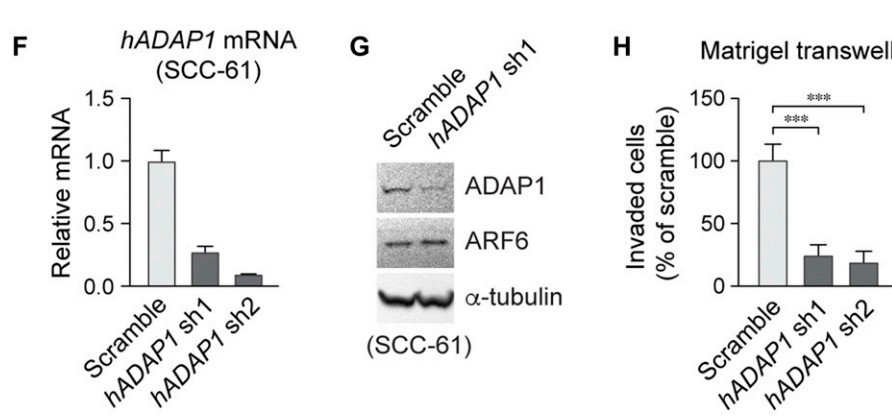

**Figure 3.  ADAP1 overexpression promotes cell invasion in vitro via its GAP activity for ARF6.**
**(A)** Quantification of invaded cells through Matrigel-coated transwell. ADAP1-WT-expressing cells showed increased invasive capability compared with control and ADAP1-R49K-expressing cells. *n* = 3–4. ***P < 0.001. *P* value by unpaired *t* test is indicated. **(B)** RT-qPCR analysis of *Arf6* mRNA in cells transduced with lentiviral *Arf6* or scramble shRNAs. *n* = 3. **(C)** Quantification of cell invasion. The ADAP1-induced invasion was reduced by *Arf6* knockdown. *n* = 4. ***P < 0.001. *P* value by unpaired *t* test is indicated. **(D)** Western blot analysis of cell lysates overexpressing ARF6-WT, ARF6-CA, and ADAP1. α-tubulin, loading control. **(E)** Quantification of invaded cells. ADAP1-induced cell invasion was reduced by ARF6-CA expression. *n* = 4. *P < 0.05. *P* value by unpaired *t* test is indicated. **(F)** RT-qPCR analysis of human *ADAP1* (*hADAP1*) RNA in SCC-61 cells transduced with lentiviral *hADAP1* or scramble shRNAs. *n* = 3. **(G)** Western blot analysis of SCC-61 cell lysates transduced with scramble or *hADAP1* shRNAs. **(H)** Quantification of invaded cells. Invasive activity of SCC-61 cells was reduced by *hADAP1* knockdown. *n* = 3. Data are mean ± SD. ***P < 0.001, *P < 0.05. *P* value by unpaired *t* test is indicated. ns, not significant.

active (CA), that is, a GTPase-deficient, mutant of ARF6 (ARF6-Q67L) to prevent the proper cycling between the GTP- and GDP-bound forms (Fig 3D). Consistent with the previous studies showing that the ARF6-CA mutant suppresses the invasive activity of cancer cell lines (Hashimoto et al, 2004; Knizhnik et al, 2012), forced expression of the ARF6-CA mutant in ADAP1-overexpressing cells reduced the invasive activity, whereas ARF6-WT did not affect (Fig 3E). These results support the idea that ADAP1 promotes cell invasion by stimulating the GTPase activity of ARF6 and facilitating the cycling between ARF6-GTP and ARF6-GDP.

We also examined the role of ADAP1 in human cancer cell invasion in vitro. In an invasive HNSCC cell line, SCC-61 (Quintavalle et al, 2011), *ADAP1* mRNA expression was efficiently suppressed lentiviral shRNAs against human *ADAP1* (Fig 3F). Unlike primary mouse keratinocytes, SCC-61 cells expressed detectable levels of ADAP1 protein, which was depleted by *ADAP1* shRNA (Fig 3G). And importantly, ADAP1 depletion significantly suppressed the invasive activity of SCC-61 cells (Fig 3H). Together, these results indicate that elevated expression of ADAP1 promotes invasion both in murine and human cells in vitro.

### ADAP1 overexpression in vivo induces invasive SCC with disrupted BMs

Our data suggest that the GAP activity-dependent and GAP activity-independent functions of ADAP1 synergistically promote invasive cell migration, which could be the potential link between high ADAP1 expression and poor clinical outcomes in patients. To investigate the impact of ADAP1 up-regulation on SCC development in vivo, we prepared lentiviral vectors that encode rtTA and Cre recombinase, and ADAP1 under the *TetO* promoter (Fig 4A). Through in utero lentivirus injection (Beronja et al, 2010; Oshimori et al, 2015), we transduced epidermal progenitor cells of *Rosa26-Lox-Stop-Lox-YFP, TetO-Hras* transgenic mouse embryos (Fig 4B). In postnatal mice, lentivirally transduced skin keratinocytes express YFP, in which the oncogenic HRAS$^{G12V}$ transgene and ADAP1 are expressed in a Dox-dependent manner to induce tumor formation.

Overall tumor morphology assessed by hematoxylin and eosin (H&E) showed that ADAP1-overexpressing tumors were highly invasive (Fig 4C), in which a significant ADAP1 expression was confirmed in tumor epithelial cells (Fig 4D). We then assessed the integrity of the BM by immunofluorescence of laminin-332, the primary laminin type of the BM in SCC (Marinkovich, 2007). Control tumors typically showed mixed phenotypes, that is, intact and breached BMs (Fig 4E, insets 1 versus 2). In contrast, ADAP1-overexpressing tumors mostly lost the BM-like laminin-332 staining pattern. Interestingly, ADAP1-overexpressing, YFP$^+$ tumor cells often exhibited laminin-332 in the cytoplasmic space (Fig 4E, insets 3 and 4). The cytoplasmic localization of laminin was not an artifact of ADAP1 overexpression because the intracellular laminin signal was also detected in invasive areas of control tumors, where the BM-like laminin-332 was broken down (Fig 4E, inset 2). Moreover, in ADAP1-overexpressing tumors, the other major component of the BM, type IV collagen, was almost undetectable at tumor-stroma interface, whereas collagen staining surrounding the vasculature remained (Fig 4F, middle).

We next asked whether the GAP activity of ADAP1 is involved in invasive tumor formation in vivo. To address this, we overexpressed

the ADAP1-R49K mutant in tumor cells. Interestingly, ADAP1-R49K overexpression resulted in the formation of tumors with thin finger-like downgrowths somewhat similar to ADAP1-WT-overexpressing tumors (Fig 4F, right). However, immunofluorescence analysis revealed a striking difference in BM composition between the two groups (Fig 4F, middle versus right). In stark contrast to the ADAP1-WT counterpart, ADAP1-R49K–overexpressing tumors maintained the BM-like staining pattern of both laminin and collagen in otherwise morphologically invasive areas (Fig 4F, right). Furthermore, we evaluated the invasiveness by quantifying the number of single-cell, two-cell, and three-cell clusters of YFP$^+$ tumor cells in the adjacent stromal areas. Compared with controls, ADAP1-WT-overexpressing tumors had more individually invasive cells, which was not the case in ADAP1-R49K-overexpressing tumors (Fig 4G). Consistent with our in vitro data, these results suggest that ADAP1 plays both GAP activity-dependent and GAP activity-independent roles in invasive tumor growth in vivo.

### TGF-$\beta$-responding tumor cells show cytoplasmic laminin, which is associated with BM breakdown

TGF-$\beta$-responding tumor cells up-regulated ADAP1 at the tumor-stroma interface (Fig 1C and D), which raised the possibility that ADAP1 may activate a molecular pathway that underlies TGF-$\beta$-induced invasive tumor progression. Therefore, we sought to determine the relevance of TGF-$\beta$ signaling to the integrity of the BM in SCC. In tumor-adjacent tissue, which does not have significant TGF-$\beta$ signaling activity (Oshimori et al, 2015), laminin-332 and type IV collagen completely overlapped at the epidermal-dermal interface demarcated by integrin $\alpha$6 staining (Fig 5A). Using a lentiviral vector containing a TGF-$\beta$ signaling reporter (Fig 5B), we examined the relevance of TGF-$\beta$ signaling activity in laminin localization at pre-invasive tumor areas. Whereas TGF-$\beta$ reporter-low tumor fronts maintained laminin-positive BM, we found that regions with TGF-$\beta$ reporter-positive cells often lost BM-like laminin staining (Fig 5C). Moreover, quantification of the intracellular laminin signal indicated that TGF-$\beta$-responding tumor cells significantly accumulated laminin in their cytoplasm (Fig 5D). Although SCC tumor cells are known to produce laminin-332 proteins (Marinkovich, 2007), the expression levels of all laminin-332 genes (*Lama3, Lamb3,* and *Lamc2*) were similar between TGF-$\beta$-responding and non-responding tumor cells in vivo (Fig 5E). These results suggest that the cytoplasmic laminin observed in TGF-$\beta$-responding cells may have been internalized from the BM. In support of this, the internalization of laminin was reported in murine mammary glands as well as human cancer cell lines (Leonoudakis et al, 2014; Muranen et al, 2017). We then examined whether this potential laminin internalization is associated with the mislocalization of type IV collagen. The BM collagen disappeared as laminin localization transitioned from the BM to the cytoplasmic space of tumor cells (Fig 5F). Furthermore, tumor areas with active TGF-$\beta$ signaling lost collagen$^+$ BM at the tumor-stroma interface, whereas regions with low reporter activity maintained BM collagen localization (Fig 5G, insets 1 versus 2). These results suggest that TGF-$\beta$-responding tumor cells triggered the breakdown of the BM potentially by internalizing laminin.

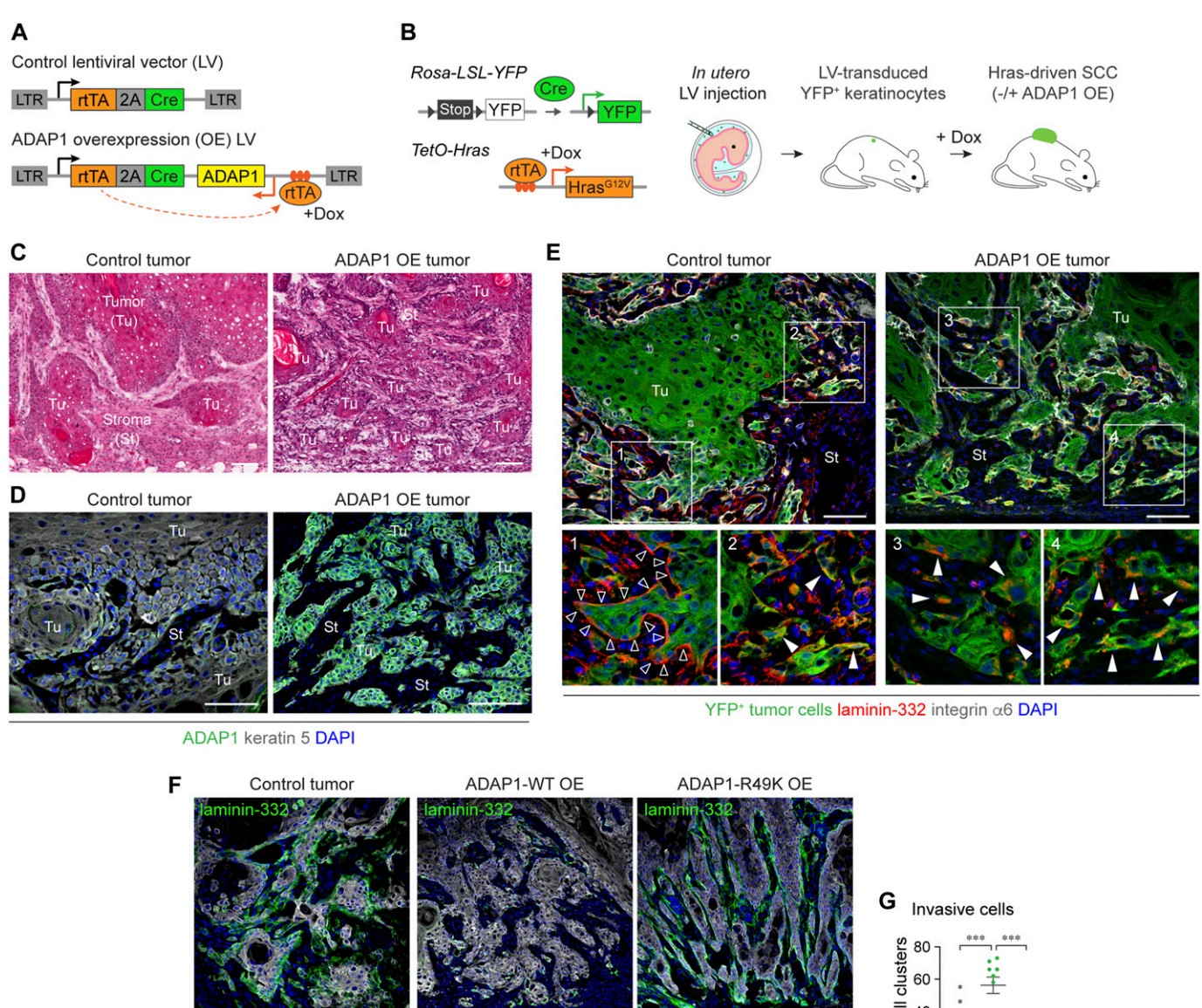

**Figure 4. ADAP1 overexpression in vivo induces invasive SCC with disrupted BMs.**
**(A)** Lentiviral vector (LV) constructs for ADAP1 overexpression. **(B)** Mouse model of SCC via LV transduction of epidermal progenitor cells of *TetO-Hras*; *Rosa-LSL-YFP* embryos by in utero microinjection. **(C)** H&E staining of tumor sections shows that smoother edges and more enclosed tumor areas in control HRAS-driven tumors. In contrast, ADAP1-WT-overexpressing tumors have multiple small and less enclosed tumor areas, suggesting that ADAP1-overexpressing tumors have an increased invasive capacity. **(D)** Immunofluorescence detection of ADAP1 protein in control and ADAP1-overexpressing tumors. **(E)** Immunolabeling of laminin-332 in tumor sections. YFP expression indicates lentivirally transduced tumor epithelial cells. Laminin-332 of the BM (black arrowheads, inset 1) was detected in control tumors (left) but is disrupted in ADAP1-overexpressing tumors (right). In the invasive regions of YFP+ tumor cells show cytoplasmic laminin-332 signal, which is more evident in ADAP1-overexpressing tumors (white arrowheads, insets 2–4). **(F)** Co-immunolabeling of laminin-332 and type IV collagen (collagen IV) in control (left), ADAP1-WT- (middle), and ADAP1-R49K-overexpressing tumors (right). Both laminin-332 (green) and collagen IV (red) are disrupted in ADAP1-WT-overexpressing tumors but largely remained in those of ADAP1-R49K-overexpressing tumors. Note: ADAP1-WT tumors show smaller pieces of tissue breaking off of the main tumor, whereas ADAP1-R49K tumors do not. Asterisks denote collagen IV staining from the endothelial vasculature that is not part of the epithelial BM. **(G)** Quantification of single cells and clusters of 2–3 cells individually invading the surrounding stroma. *n* = 3–5. Data are shown in scatter plots with mean ± SEM. ***P < 0.001. *P* value by unpaired *t* test is indicated. Scale bars, 50 μm.

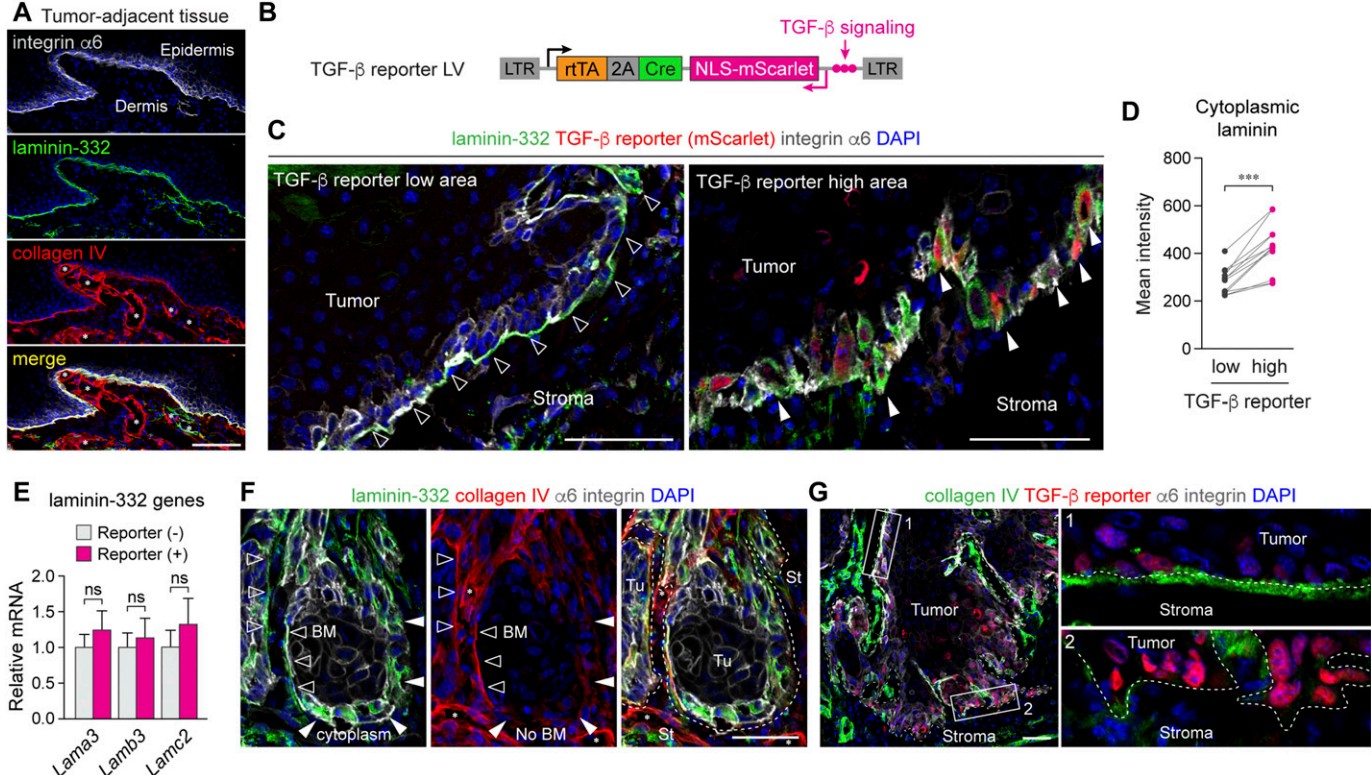

**Figure 5. TGF-β-responding tumor cells show cytoplasmic laminin, which is associated with BM breakdown.**
**(A)** Immunodetection of BM components at the epidermal-dermal interface in tumor-adjacent tissue. **(B)** LV construct containing TGF-β reporter for SCC formation.
**(C)** Immunolabeling of laminin-332 at the tumor-stroma interface in TGF-β reporter low (left) and TGF-β reporter high (right) regions. In reporter low regions, the BM detected by laminin-332 is intact (black arrowheads), whereas in reporter high regions, laminin is detected in cytoplasmic space of TGF-β reporter⁺ cells and BM-like laminin staining is disappeared (white arrowheads). **(D)** Quantification of cytoplasmic laminin localization in TGF-β reporter low and high tumor basal cells. n = 3–4, total 12 areas. ***P < 0.001. P value by paired t test is indicated. **(E)** Relative mRNA expression of laminin-332 genes (Lama3, α3 subunit; Lamb3, β3 subunit; Lamc2, γ2 subunit) in TGF-β reporter (−) and (+) tumor basal cells isolated by FACS and analyzed by RT-qPCR. Data are mean ± SEM. ns, not significant. **(F)** Co-immunolabeling of laminin-332 (green) and collagen IV (red) at the tumor-stroma interface. Note, tumor regions with BM-like staining of laminin-332 maintain collagen IV BM staining (black arrowheads), whereas tumor regions showing cytoplasmic laminin-332 staining lose BM-like collagen staining (white arrowheads). Collagen IV staining of the tumor vasculature remains intact (asterisks).
**(G)** Immunolabeling of collagen IV at the tumor-stroma interface. Dotted lines denote the tumor-stroma interface based on integrin α6 staining. Note, the BM detected by collagen IV remains in TGF-β reporter low regions (inset 1) but are disrupted in TGF-β reporter high regions (inset 2). Scale bars, 50 μm.

## ADAP1 depletion reduces cytoplasmic laminin localization in TGF-β-responding tumor cells and suppresses invasive tumor growth

To determine whether ADAP1 is involved in invasive tumor growth and laminin internalization by TGF-β-responding tumor cells, we established an *Adap1*-floxed mouse line. We induced the conditional KO (cKO) of the *Adap1* gene by transducing lentiviral Cre. Successful cKO was confirmed by quantitative PCR (qPCR) and immunoblotting in vitro (Fig 6A and B). To delete the *Adap1* gene in tumor cells in vivo, we performed in utero injection of the TGF-β reporter lentiviral vectors in *Adap1*ᶠˡ/ᶠˡ mice, and in *Adap1*⁺/ᶠˡ or *Adap1*⁺/⁺ mice as controls (Fig 6C). HRASᴳ¹²ⱽ-driven tumors developing on *Adap1* cKO mice were significantly smaller than those of control mice (Fig 6D). Decreased size in *Adap1* cKO tumors suggested that ADAP1 might break a barrier for restricting extensive tumor growth. TGF-β-responding cells in *Adap1* cKO tumors still showed more cytoplasmic laminin signals than those in non-responding cells (Fig 6E). However, the quantification data revealed that TGF-β-responding tumor cells in *Adap1* cKO tumors had significantly less cytoplasmic laminin compared with TGF-β-responding

cells in control tumors (Fig 6E). Likely because of reduced laminin disruption, *Adap1* cKO tumors showed well-preserved BM collagen staining (Fig 6F) and had significantly fewer individually invading cells than those in control tumors (Fig 6G). Although other ARF6 GAPs and guanine nucleotide-exchanging factors (GEFs) might also regulate the potential laminin internalization, these results indicate that ADAP1 promotes invasive migration of TGF-β-responding tumor cells by facilitating the breakdown of the BM.

## Discussion

New diagnostic modalities to identify high-risk SCC from most low-risk tumors are urgently needed. Given that long-lived, stem-like TICs maintain and promote tumorigenesis, the mechanisms by which TICs acquire invasive properties may hold potential targets for novel cancer diagnostics and treatment. Invasive properties are likely regulated by cells surrounding TICs through modulating local ECM proteins and providing growth factor signaling. We previously

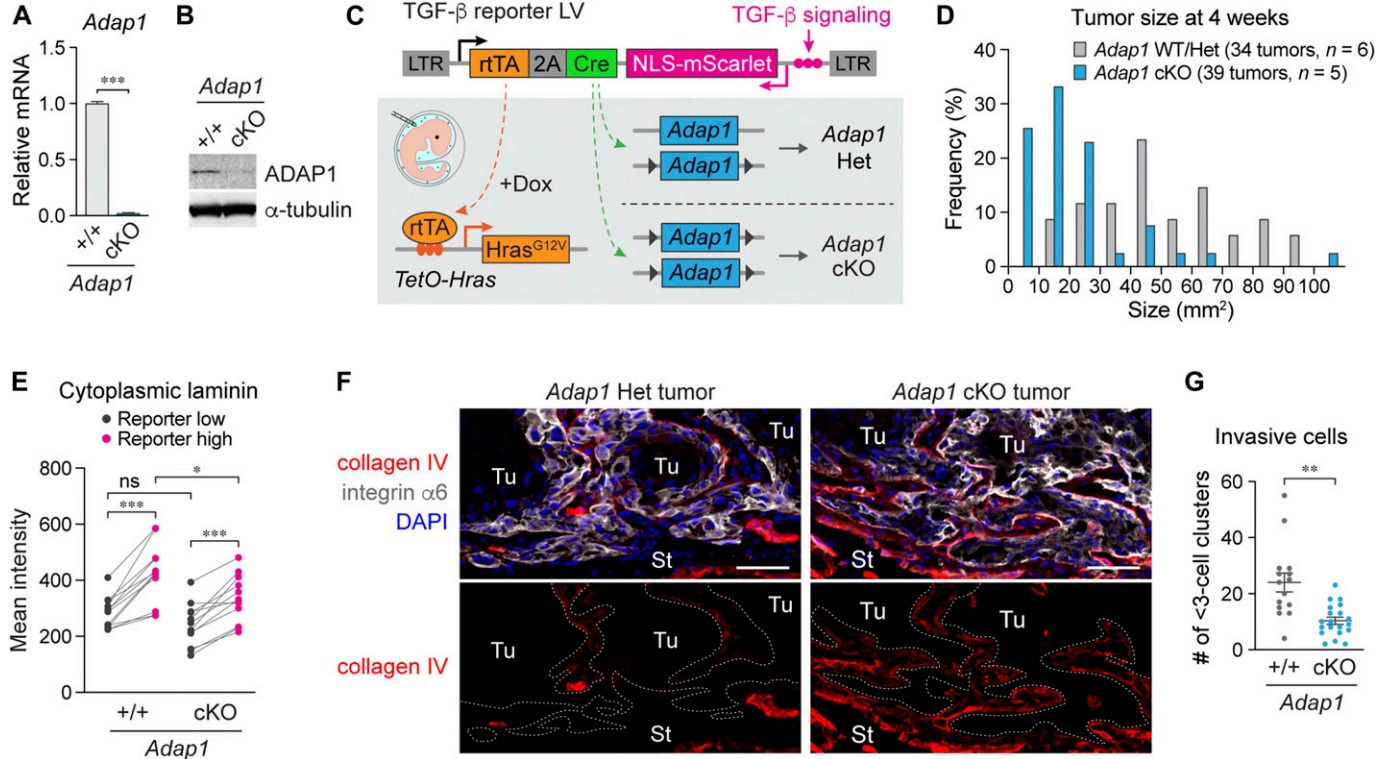

**Figure 6.  ADAP1 depletion reduces cytoplasmic laminin localization in TGF-β-responding tumor cells and suppresses invasive tumor growth.**
**(A)** RT-qPCR validation of *Adap1* deletion using primary keratinocytes from *Adap1*-floxed mice that were transduced with LVs in vitro. *n* = 3, Data are mean ± SEM. ***P < 0.001. *P* value by unpaired *t* test is indicated. **(B)** Western blot analysis of *Adap1* cKO keratinocytes and a littermate control. To detect endogenous ADAP1 protein, we used immunoprecipitated samples using the anti-ADAP1 antibody. **(C)** Schematic of in utero injection of TGF-β reporter LV to *TetO-Hras; Adap1*-floxed mice to induce a tumor-specific *Adap1* cKO. **(D)** The size of tumors developing in *Adap1* cKO mice and their WT/Het littermates were shown. Note: *Adap1* cKO mice form smaller tumors compared with the controls at 4 wk after Dox administration. **(E)** Quantification of cytoplasmic laminin intensity in TGF-β reporter low and high tumor basal cells. Intensity data from individual images are shown as dots, and lines between dots show the pairing data from each microscopic image. *n* = 3–4, total 12 areas. *P = 0.0314 (unpaired *t* test), ***P < 0.001 (paired *t* test). **(F)** Immunolabeling of collagen IV at the tumor-stroma interface. Note, BM-like staining in *Adap1* Het tumor is diminished but remains consistently at the tumor-stroma interface detected by integrin α6 staining in *Adap1* cKO tumor. Scale bars, 50 μm. **(G)** Quantification of single cells and clusters of 2–3 cells individually invading the surrounding stroma. *n* = 4–6. Data are shown in scatter plots with mean ± SEM. **P = 0.001. *P* value by unpaired *t* test is indicated. ns, not significant.

demonstrated that TGF-β-responding tumor cells are drug-resistant TICs that drive invasive SCC progression (Oshimori et al, 2015). From genes up-regulated in TGF-β-responding TICs, we identified *ADAP1* as a strong predictor of poor survival in SCC patients and, therefore, could be an excellent marker to stratify the risk of life-threatening SCC. Moreover, our functional studies demonstrated that ADAP1 promotes invasive SCC progression through cell migration and BM modification.

Through our in vitro and in vivo studies, we examined how ADAP1 is involved in SCC progression. We first showed that ADAP1 enhances cell migration in its GAP activity-independent molecular function, potentially through the regulation of the actin cytoskeleton (Thacker et al, 2004; Venkateswarlu et al, 2004). Then, we found that ADAP1 also promotes invasion on a cell-by-cell basis in vitro and, more importantly, invasive tumor growth on a tissue basis in vivo. Moreover, based on the morphology of ADAP1-R49K-overexpressing tumors, a GAP-independent function of ADAP1 seemed to be sufficient to cause finger-like epithelial downgrowths likely through the regulation of cell migration. However, as the BM remained undisrupted in those tumors, the GAP activity of ADAP1 seems necessary for the breakdown of the BM. These findings highlight that the GAP

activity-dependent and GAP activity-independent roles of ADAP1 synergistically facilitate the progression of SCC to invasive states.

The degradation of BM is essential for the transition of precursor lesions to invasive SCC. We revealed that ADAP1-rich, TGF-β-responding tumor cells had laminin-332 in the cytoplasmic space, and simultaneously lost the BM composed of laminin and type IV collagen. As a potential mechanism of BM breakdown, ADAP1 may facilitate the laminin internalization through the ARF6-mediated endocytosis. However, our data do not exclude the possibility of other ARF6's functions, including signal transduction, cell division, and cell-cell adhesion by regulating endocytic vesicle trafficking of membrane proteins (D'Souza-Schorey & Chavrier, 2006). Moreover, an ARF6-mediated site-directed targeting of matrix metalloprotease MT1-MMP also facilitates tumor cell invasion (Marchesin et al, 2015). In the future, molecular interplays coordinated by the ADAP1-ARF6 system might lead to a deep mechanistic understanding of invasive tumor progression. Although the conversion of small GTPase proteins to the GDP-bound form was traditionally considered as negative regulation, recent studies have highlighted the importance of GAPs in the engagement of ARF6 in diverse physiological contexts. Moreover, the literature suggests that instead of this

hydrolysis acting as a negative regulator of ARF6, endocytosis is dependent on the cycling of ARF6 into the GDP-bound form (Klein et al, 2006). We and others showed that the constitutively active (i.e., the GTP-hydrolysis activity defective) mutant of ARF6 had a suppressive effect on cell invasion in vitro (Hashimoto et al, 2004; Knizhnik et al, 2012).

Elevated levels of laminin-332 expression in SCC have been associated with invasiveness and poor patient clinical prognosis (Ono et al, 1999; Hamasaki et al, 2011). However, it is noteworthy that elevated laminin-332 expression was observed in the tumor cell cytoplasm (Ono et al, 1999). The mechanism by which tumor cells internalize laminin from the BM remains largely unknown. Proteolytic cleavage of laminin-332 seems to involve multiple enzymes produced by tumor cells as well as by cells in the stroma (Marinkovich, 2007). Also, integrins are involved in diverse cellular processes crucial for cancer development (Hamidi & Ivaska, 2018). In particular, $\alpha6\beta4$ integrin, a laminin-binding integrin complex, has been shown to play an essential role in SCC tumorigenesis (Janes & Watt, 2006). By binding to laminin, integrins induce signals for cell proliferation, survival, migration, and invasion (Desgrosellier & Cheresh, 2010). Moreover, integrins are internalized in response to growth factors (Ning et al, 2007) and targeted to early endosomes but not to lysosomes (Das et al, 2017), suggesting that internalized integrins are recycled to play a role at new locations. Elucidating the fate and function of internalized laminin-binding integrins and the precise molecular mechanism of ADAP1-mediated laminin internalization may provide further insight into invasive tumor growth.

## Materials and Methods

### In silico analysis of genes up-regulated in TGF-$\beta$-responding TICs using TCGA data

Among 663 genes up-regulated in TGF-$\beta$-responding tumor cells from our mouse model analyzed by RNA-seq previously (log$_2$ fold-change > 1, adj $P$ < 0.05 [Oshimori et al, 2015]), we identified 403 annotated genes in the TCGA database. From them, we removed genes with fragments per kilobase million < 5 in the RNA-seq, and genes less expressed in tumors compared with normal tissue samples (fold-change < 0.5, adj $P$ < 0.05, Mann–Whitney) in the TCGA head & neck SCC (HNSCC) database. Gene expression data of the remaining 292 genes were downloaded from TCGA HNSCC mRNA expression (Illumina-HiSeq) ($n$ = 546) via the University of California, Santa Cruz cancer browser at https://genome-cancer.ucsc.edu (data obtained on April 24th, 2016). Cases without available information of clinical stage and overall survival were excluded, and a total of 519 cases, including 118 early-stage cases, were analyzed. Multivariate Cox proportional hazards model was applied to estimate the relationship between gene expression and overall survival with adjustments by age, gender, and clinical stage, using R version 3.2.3 and R package "survival." $P$ values were adjusted for multiple comparisons using Benjamin–Hochberg false discovery rate adjustments.

### Lentivirus vector construction, production, and concentration

For the generation of an LV-rtTA-2A-Cre construct, we replaced the puromycin cassette of pLKO.1 between BamHI and NsiI sites with an open reading frame containing synthesized cDNA of the reverse tetracycline transactivator (rtTA3), a P2A linker sequence, and a PCR-amplified cDNA of codon-improved Cre recombinase (Cre). To generate an LV-rtTA-2A-Cre-TGF-$\beta$ reporter construct, we created a multi-cloning site sequence containing PmeI-NheI-BmtI-XbaI-SalI-MluI sites between NsiI and KpnI, and then inserted a DNA fragment containing 12× repeated SMAD-binding elements (SBE, 5'-AGCCA-GACA-3'), a 113-bp minimal CMV promoter, mScarlet-I, and cHS4 insulator sequences between KpnI and NheI sites. To improve the sensitivity and transgene expression, we upgraded the construct with a brighter version of nuclear red fluorescent protein, NLS-mScarlet-I (Bindels et al, 2017) (plasmid ID 85044; Addgene), and inserted the chicken hypersensitivity site 4 insulator (Ins) (Aker et al, 2007). ADAP1 arginine 49 to lysine (R49K) mutation was introduced by PCR using primers containing two nucleotide mutations and a BstZ17I restriction enzyme site. The mutation was confirmed by Sanger sequencing. To generate a doxycycline-inducible ADAP1 expression system, we first generated LVs that constitutively express hygromycin resistance (HygR) gene under the phosphoglycerate kinase (PGK) promoter and conditionally express NLS-mRFP from the tetracycline-response elements (TetO promoter) in a doxycycline (Dox)-dependent manner and then, the second LVs that constitutively express rtTA and conditionally express ADAP1-WT or ADAP1-R49K mutant in a Dox-dependent manner. For generating ARF6-expression lentiviral vectors, we obtained pcDNA3-Arf6 (WT) and Arf6 (Q67L) from Addgene (plasmid ID 79424 and 79425, respectively) and subcloned PCR-amplified Arf6 fragments into pLKO vector, which was followed by sequence confirmation. DNA plasmid mini/midi/maxi-prep, gel extraction, and PCR clean-up kits from Zymo Research were used. Large-scale production of vesicular stomatitis virus-G pseudotyped lentivirus was performed by calcium phosphate transduction of 293T cells (632180; Clontech/Takara) plated on poly-L-lysine–coated cell-culture dishes with pLKO.1 and helper plasmids: pMD2.G and psPAX2 (plasmid 12259 and 12260; Addgene, respectively). Viral supernatant was collected 30–36 h after transfection, filtered through a 0.45-$\mu$m polyvinylidene difluoride (PVDF) filter (Millipore), concentrated first by passage through a Vivacell 70 centrifugal concentrator (100 kD molecular weight cutoff, VS6041; Sartorius), and then by ultracentrifugation in an Ultra-Clear Tube (344057; Beckman Coulter) for 90 min in an MLS-50 rotor at 45,000 rpm or an SW-55 Ti rotor at 42,000 rpm (Beckman Coulter).

### In utero ultrasound-guided lentivirus microinjection and tumor formation

The following mice were used: *Tg(TetO-HRAS)65Lc/Nci* (*TetO-Hras$^{G12V}$*, NCI Mouse Repository, donated by L Chin) (Chin et al, 1999) and *Gt(ROSA)26Sor$^{tm(EYFP)Cos}$/J* (*Rosa26-LSL-EYFP*, The Jackson Laboratory, donated by F Costantini) (Srinivas et al, 2001). To generate *Adap1*-floxed mice, we obtained cryopreserved sperm of *Adap1$^{tm1a(EUCOMM)Wtsi}$* from the European Conditional Mouse Mutagenesis consortium, re-derived the mutant mice, and crossed with an FLPo mouse (*Tg(Pgk1-flpo)10Sykr/J*). The FLPo-mediated deletions of the *lacZ* and *neo* cassettes flanked by FRT sequences were confirmed by genotyping PCR, which indicates the "conditional ready" floxed allele. In utero ultrasound-guided microinjection was performed as previously described (Beronja et al, 2010). Briefly, females at E9.5 of gestation were

anesthetized with isoflurane on the heated stage before injection. 300–500 nl of lentivirus suspension was injected into each embryo's amniotic sac. Surgical procedures were limited to 30 min to maintain high survival rates. Postnatally, the lentivirally transduced mice were given mouse chow containing doxycycline (2 mg/kg, 5AKR; TestDiet), which activates rtTA to induce the expression of oncogenic HRAS[G12V] or HRAS[G12V] and ADAP1 simultaneously. The Oregon Health & Science University Animal Care and Use Committee approved animal experimentation protocols used in this study.

### Tissue harvest, sectioning, and immunofluorescence microscope imaging

Because of the humane endpoints based on tumor size, number, and location of the body surface, mice were euthanized at different time points after doxycycline-induced HRAS[G12V] expression. Before tumor collection, dorsal hairs were removed with either an electric hair clipper or hair removal cream. Mice were then photographed, and tumor size was measured with a ruler. Tumors were dissected and fixed with 4% paraformaldehyde (16% solution; Electron Microscopy Sciences) in PBS for 15 min at room temperature. After washing with PBS overnight, tumor tissues were embedded and frozen in Tissue-Tek optimum cutting temperature compound (4583; Sakura) in a cryomold (4557; Sakura). 5–10 $\mu$m sections were cut with a Cryostat (CM1850; Leica), mounted on SuperFrost Plus slides (Fisherbrand), and permeabilized for 30 min in 0.3% Triton X-100 in PBS. For tyramide signal amplification Plus kit amplification (NEL744001KT; PerkinElmer), and to remove background fluorescence, the tissues were instead treated with 1% hydrogen peroxide ($H_2O_2$) in 0.3% Triton X-100/PBS for 30 min at room temperature. When immunolabeling with mouse antibodies, the sections were incubated with the Mouse-On-Mouse blocking kit according to the manufacturer's instructions (BMK-2202; Vector Laboratories). Fresh frozen cutaneous SCC patient optimum cutting temperature samples from the Oregon Health & Science University Dermatology department were similarly immunolabeled. The following primary antibodies were used: rabbit anti-ADAP1/centaurin-$\alpha$1 (ab197380; Abcam), mouse anti-SMAD2/3 (610842; BD Biosciences), rat anti-$\alpha$6 integrin (GoH3; BD Biosciences), rabbit anti-mCherry (ab167453; Abcam), goat anti-mCherry (AB0040; Acris), anti-chicken anti-GFP (to detect YFP) (ab13970; Abcam), rabbit anti-laminin 5/laminin-332 (ab14509; Abcam), goat anti-type IV collagen (AB769; Millipore), chicken anti-keratin 5 (SIG-3475; Covance), Alexa Fluor 647–conjugated phalloidin (A22287; Thermo Fisher Scientific), and Alexa Fluor 647–conjugated WGA (W32466; Thermo Fisher Scientific). Sections were treated with primary antibody mixes and incubated at 4°C for overnight. After washing with PBS containing 0.1% Triton X-100 (PBST), the sections were treated for 30 min at room temperature with secondary antibodies conjugated with Alexa Fluor 488, 546, or 647 (Jackson ImmunoResearch and Thermo Fisher Scientific). Slides were washed, counterstained with DAPI, and mounted in Prolong Gold (Thermo Fisher Scientific).

### In vitro cell culture experiments

Keratinocytes isolated from neonatal mouse epidermis were maintained in E medium with 10% FBS and 50 $\mu$M $CaCl_2$ at 37°C, 7.5%

$CO_2$. Human head and neck cell lines, SCC-61, were maintained in DMEM with 10% FBS. For lentiviral transduction in vitro, cells were plated in six-well plates at $1.0 \times 10^5$ cells per well and incubated with viruses in the presence of polybrene (10 $\mu$g/ml) for 30 min at 37°C, 7.5% $CO_2$, and then the plates were spun at 1,100$g$ for 30 min at 37°C in an Eppendorf 5810R centrifuge. After centrifugation, virus-containing medium was removed and replaced with fresh growth medium. For establishing the ADAP1 overexpression system, the cells were transduced with the first LV, which express HygR gene from the PGK promoter and NLS-mRFP from the *TetO* promoter. After hygromycin selection, the cells were transduced with the second LV that expresses rtTA from the PGK promoter and ADAP1 from the *TetO* promoter. For the rtTA-mediated inducible gene expression, the medium was supplemented with Dox (100 ng/ml). For TGF-$\beta$ stimulation, the medium was supplemented with recombinant murine TGF-$\beta$1 (100 pM, 7666-MB; R&D systems). For immunolabeling and visualization, the cells were seeded onto coverslips and fixed with pre-warmed 4% PFA for 10 min at 37°C. The cells were permeabilized with 0.3% Triton X-100 in PBS and incubated at 4°C overnight with the primary antibodies listed above. After washing with PBST, the sections were treated for 30 min at room temperature with secondary antibodies conjugated with Alexa Fluor 488, 546, or 647. Slides were washed, counterstained with DAPI, and mounted in Prolong Gold. Only for cell surface protein detection, non-permeabilized cells were incubated with primary and secondary antibodies followed by Hoechst 33342 (H3570; Thermo Fisher Scientific) counterstaining.

### Image acquisition and analysis

Imaging was performed on a Zeiss AxioObserver.Z1 equipped with Apotome.2 using ×10/0.45, ×20/0.8, or ×40/0.95 air. Images were collected using the Zeiss Efficient Navigation software and processed using an image processing package, Fiji (Schindelin et al, 2012; Rueden et al, 2017). For laminin internalization analysis, a rolling-ball background subtraction of 50 pixels was used on all channels. Directional filtering (20-pixel line length in 32 directions via the erosion method) of the keratin-5 channel was then conducted using the MorphoLibJ package (Legland et al, 2016). Individual cells were then selected from the filtered keratin-5 channel with MorphoLibJ Morphological Watershed Segmentation, with tolerance ranges from 100 to 500, connectivity of 8, and dams calculated (Legland et al, 2016). Labels not in contact with the BM were removed by hand, so that a single cell layer remained. Intensity was then measured for laminin, DAPI, and mScarlet channels via MorphoLibJ 2D Intensity analysis (Legland et al, 2016). Further processing and statistics were conducted in RStudio v1.2.1335 (RStudio Team, 2015). The intensity of the reporter channel of each label in an image was divided by the maximum reporter intensity for that image, creating the "reporter ratio" for each label. Based on density plots, a threshold was determined, and all labels with a reporter ratio above 0.2 were classified as "reporter high" (around 10% of all cells), whereas all labels with a reporter ratio below 0.2 were classified as "reporter low."

### RNA purification and RT-qPCR

Total RNA from cells in TRI Reagent (Zymo Research) was purified using Direct-zol RNA MicroPrep Kit (Zymo Research) as per the manufacturer's instructions. For RT-qPCR, an equivalent amount of

RNA was reverse-transcribed by SuperScript IV VILO cDNA synthesis Kit (11756050; Invitrogen). cDNAs were mixed with indicated primers and PowerUp SYBR Green Master Mix (A25742; Applied Biosystems), and RT-qPCR was performed on a ViiA 7 Real-Time PCR System (Applied Biosystems). cDNAs were normalized to equal amounts using primers against *Actb*. The following primer sequences were used (5′-3′):

*Actb* forward: GGCTGTATTCCCCTCCATCG, *Actb* reverse: CCAGTTGGTAACAATGCCATGT;
*Adap1* forward: GCCCAAGTTGAGTTCATGGC, *Adap1* reverse: GTAGAAGGGCGGTACTTTGGA;
*Arf6* forward: TTCGGGAACAAGGAAATGCG, *Arf6* reverse: GGATGGTGGTCACCGATTGG;
*Lama3* forward: CTGTGACTACTGCAATTCTGAGG, *Lama3* reverse: CAAGGTGAGGTTGACTTGATTGT;
*Lamb3* forward: GGCTGCCTCGAAATTACAACA, *Lamb3* reverse: ACCCTCCATGTCTTGCCAAAG;
*Lamc2* forward: CAGACACGGGAGATTGCTACT, *Lamc2* reverse: CCACGTTCCCCAAAGGGAT;
*ADAP1* forward: CGCGAGAGCCAGGTTTGAG, *ADAP1* reverse: GCTTCTCCGGGTAGATGAACT.

### shRNA sequences for gene knockdown

For mouse *Arf6* gene and human *ADAP1* gene knockdown experiments, DNA oligos of following target sequences were synthesized (5′-3′):

the scramble shRNA, CAACAAGATGAAGAGCACCAA;
*Arf6* sh4, CCGGAAGGAGAGAAATCCAAA;
*Arf6* sh7, GCATTACTACACCGGGACCCA;
*ADAP1* sh1, CCCACCTCCACGACTATTTAT;
*ADAP1* sh2, TCGGGAATCCACCGGAATATC.

### Protein gel electrophoresis and immunoblotting

Total cell lysates were prepared using RIPA (20 mM Tris–HCl, pH 8.0, 150 mM NaCl, 1 mM EDTA, 1 mM EGTA, 1% Triton X-100, 0.5% deoxycholate, and 0.1% SDS) supplemented with protease inhibitors (Halt Protease Inhibitor, 78430; Thermo Fisher Scientific), or used directly with 1X Bolt LDS Sample Buffer (B0007; Invitrogen). The protein concentrations of clarified supernatants were measured by a BCA Protein Assay Kit (23227; Pierce). Immunoprecipitation of endogenous ADAP1 was performed by using 5 $\mu$l of Dynabeads Protein G (10003D; Thermo Fisher Scientific) bound with 1 $\mu$g goat anti-ADAP1/Centaurin-$\alpha$1 (ab27476; Abcam) antibody. Proteins in lysates and immunoprecipitated complexes were separated by Bolt 4–12% Bis-Tris Plus gels (Invitrogen) and transferred to a PVDF membrane (Immobilon-FL, IPFL00010; Millipore) by a Pierce Power Blotter XL System (PB0013; Invitrogen). Transferred membranes were blocked for 1 h in Odyssey Blocking Buffer (927-50000; LI-COR) diluted 1:1 in TBS (20 mM Tris–HCl, pH 7.4, 150 mM NaCl) and then incubated with primary antibodies in the blocking buffer or TBS overnight at 4°C. After washing with TBS, the membranes were incubated with secondary antibodies in the Odyssey Blocking Buffer diluted 1:1 in TBS for 1 h at room temperature. Membranes were washed in TBS

and imaged on an Azure c600 Infra-Red Imaging System (Azure Biosystems). Primary antibodies used were rabbit anti-ADAP1/centaurin-$\alpha$1 (ab197380; Abcam) and mouse anti-$\alpha$-tubulin (DM1A; Sigma-Aldrich). IRDye680- or IRDye800-conjugated secondary antibodies were used (1:10,000; LI-COR).

## Supplementary Information

## Acknowledgements

We thank the European Conditional Mouse Mutagenesis (EUCOMM) consortium for providing the cryo-sperm of *Adap1*[tm1a(EUCOMM)Wtsi], L Chin for *TetO-Hras* mice, and F Costantini for *Rosa-YFP* mice. We also thank S Courtneidge for providing SCC-61 cell line and comments on our manuscript. We appreciate the assistance of Oregon Health & Science University (OHSU)'s Department of Comparative Medicine (an Association for Assessment and Accreditation of Laboratory Animal Care facility) for expert care and housing of our mouse colony, and the OHSU transgenic mouse core for assisting with in vitro fertilization of *Adap1*[tm1a(EUCOMM)Wtsi] mice (L Fedorov, Director). This work is supported by the NIH K99-R00 pathway to independence award (R00-CA178197), the OHSU Knight Cancer Institute (KCI) Pilot Grant (2018-CCSG-22), and the start-up support from the KCI/OHSU to N Oshimori.

### Author Contributions

A Van Duzer: conceptualization, data curation, formal analysis, validation, investigation, visualization, methodology, and writing—original draft, review, and editing.
S Taniguchi: conceptualization, data curation, formal analysis, validation, investigation, visualization, methodology, and writing—review and editing.
A Elhance: data curation, formal analysis, visualization, and writing—review and editing.
T Tsujikawa: data curation, formal analysis, visualization, methodology, and writing—review and editing.
N Oshimori: conceptualization, data curation, formal analysis, supervision, funding acquisition, validation, investigation, visualization, methodology, project administration, and writing—original draft, review, and editing.

### Conflict of Interest Statement

The authors declare that they have no conflict of interest.

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
