## [Reviewer comments · Life Science Alliance]

ADAP1 promotes invasive squamous cell carcinoma progression and predicts patient survival

Avery Van Duzer, Sachiko Taniguchi, Ajit Elhance, Takahiro Tsujikawa, and Naoki Oshimori

DOI: 10.26508/lsa.201900582

Corresponding author(s): Dr. Naoki Oshimori (Knight Cancer Institute, Oregon Health & Science University)

Review timeline:

Submission Date:	2019-10-22
Editorial Decision:	2019-10-23
Revision Received:	2019-10-29
Editorial Decision:	2019-10-31
Revision Received:	2019-11-04
Accepted:	2019-11-05

Scientific Editor: Andrea Leibfried

Transaction Report:

No Peer Review Process File is available with this article, as the authors have chosen not to make the review process public in this case.

1st Editorial Decision

23 October 2019

October 23, 2019

Re: Life Science Alliance manuscript #LSA-2019-00582-T

Dr. Naoki Oshimori
Department of Cell, Developmental & Cancer Biology
Knight Cancer Institute
Oregon Health & Science University
2720 SW Moody Ave
KCRB 3002
Portland, OR 97201, USA

Dear Dr. Oshimori,

Thank you for transferring your manuscript entitled "Tumor-initiating cells trigger basement membrane breakdown via ADAP1 to promote invasive tumor progression" to Life Science Alliance. The manuscript was assessed by expert reviewers at another journal before, and the editors transferred those reports to us with your permission.

The reviewers appreciated the proposed role of ADAP1 for tumor invasiveness, but thought that its role in breakdown of the basement membrane was not sufficiently supported and that the interplay with Arf6 would need better insight. We think that a revised version concentrating on ADAP1 as a predictor of survival and on its role in migration in conjunction with Arf6 would be suitable for publication in Life Science Alliance, and we would thus like to invite you to submit a revised version to us. Importantly, we would expect a full point-by-point response to the reviewer concerns. Please provide better experimental support for the proposed link to Arf6 regulation underlying the observed phenotype and remove the part suggesting alterations of integrin / BM downstream of this. The requested controls, side-by-side comparison, number of patients (rev#3), and clarifications should get provided and ADAP1 effects in SCCs should get tested (rev#1 and #2).

To upload the revised version of your manuscript, please log in to your account:
<https://lsa.msubmit.net/cgi-bin/main.plex>
You will be guided to complete the submission of your revised manuscript and to fill in all necessary information. Please get in touch in case you do not know or remember your login name.

Thank you for this interesting contribution to Life Science Alliance. We are looking forward to receiving your revised manuscript.

Sincerely,

Andrea Leibfried, PhD
Executive Editor
Life Science Alliance
Meyershofstr. 1
69117 Heidelberg, Germany
t +49 6221 8891 502
e a.leibfried@life-science-alliance.org
www.life-science-alliance.org

B. MANUSCRIPT ORGANIZATION AND FORMATTING:

October 31, 2019

RE: Life Science Alliance Manuscript #LSA-2019-00582-TR

Dr. Naoki Oshimori
Knight Cancer Institute, Oregon Health & Science University
Cell, Developmental & Cancer Biology
2720 SW Moody Ave
Portland, OR 97201

Dear Dr. Oshimori,

Thank you for submitting your revised manuscript entitled "ADAP1 promotes invasive squamous cell carcinoma progression and predicts patient survival". I now assessed the revised version of your manuscript and appreciate the changes introduced and the experiments added, including those added to better link ADAP1-mediated effects to Arf6 mis-regulation. I would thus be happy to publish your paper in Life Science Alliance pending final revisions necessary to meet our formatting guidelines:

- please state in the figure legends next to the p-values which statistical test has been employed
- you mention an RNA-seq experiment for figure 5E, please correct (I assume the results refer to RT-PCR analysis as indicated in the M&M section)

A. FINAL FILES:

B. MANUSCRIPT ORGANIZATION AND FORMATTING:

Sincerely,

Andrea Leibfried, PhD
Executive Editor
Life Science Alliance
Meyershofstr. 1
69117 Heidelberg, Germany
t +49 6221 8891 502
e a.leibfried@life-science-alliance.org
www.life-science-alliance.org

3rd Editorial Decision

5 November 2019

November 5, 2019

RE: Life Science Alliance Manuscript #LSA-2019-00582-TRR

Dr. Naoki Oshimori
Knight Cancer Institute, Oregon Health & Science University
Cell, Developmental & Cancer Biology
2720 SW Moody Ave
Portland, OR 97201

Dear Dr. Oshimori,

Thank you for submitting your Research Article entitled "ADAP1 promotes invasive squamous cell carcinoma progression and predicts patient survival". It is a pleasure to let you know that your manuscript is now accepted for publication in Life Science Alliance. Congratulations on this interesting work.

DISTRIBUTION OF MATERIALS:

Again, congratulations on a very nice paper. I hope you found the review process to be constructive and are pleased with how the manuscript was handled editorially. We look forward to future exciting submissions from your lab.

Sincerely,

Andrea Leibfried, PhD

Executive Editor
Life Science Alliance
Meyerhofstr. 1
69117 Heidelberg, Germany
t +49 6221 8891 502
e a.leibfried@life-science-alliance.org
www.life-science-alliance.org